# A flexible age-dependent, spatially-stratified predictive model for the spread of COVID-19, accounting for multiple viral variants and vaccines

**Kristan Alexander Schneider**[1]☺*, **Henri Christian Junior Tsoungui Obama**[1,2]☺,
**Nessma Adil Mahmoud Yousif**[1,2]☺

**1** Department of Applied Computer- and Biosciences, University of Applied Sciences Mittweida, Mittweida, Germany, **2** African Institute for Mathematical Sciences Cameroon, Limbe, Cameroon

☺ These authors contributed equally to this work.
* kristan.schneider@hs-mittweida.de

## Abstract

### Background

After COVID-19 vaccines received approval, vaccination campaigns were launched world-wide. Initially, these were characterized by a shortage of vaccine supply, and specific risk groups were prioritized. Once supply was guaranteed and vaccination coverage saturated, the focus shifted from risk groups to anti-vaxxers, the under-aged population, and regions of low coverage. At the same time, hopes to reach herd immunity by vaccination campaigns were put into perspective by the emergence and spread of more contagious and aggressive viral variants. Particularly, concerns were raised that not all vaccines protect against the new-emerging variants. The objective of this study is to introduce a predictive model to quantify the effect of vaccination campaigns on the spread of SARS-CoV-2 viral variants.

### Methods and findings

The predictive model introduced here is a comprehensive extension of the one underlying the pandemic preparedness tool CovidSim 2.0 (http://covidsim.eu/). The model is age and spatially stratified, incorporates a finite (but arbitrary) number of different viral variants, and incorporates different vaccine products. The vaccines are allowed to differ in their vaccination schedule, vaccination rates, the onset of vaccination campaigns, and their effectiveness. These factors are also age and/or location dependent. Moreover, the effectiveness and the immunizing effect of vaccines are assumed to depend on the interaction of a given vaccine and viral variant. Importantly, vaccines are not assumed to immunize perfectly. Individuals can be immunized completely, only partially, or fail to be immunized against one or many viral variants. Not all individuals in the population are vaccinable. The model is formulated as a high-dimensional system of differential equations, which is implemented efficiently in the programming language Julia. As an example, the model was parameterized to reflect the epidemic situation in Germany until November 2021 and future dynamics of the

**Data Availability Statement:** Only simulated data is reported. The data can be reproduced by the parameters specified in the manuscript and the Julia code provided at GitHub: https://github.com/Maths-against-Malaria/COVID19_Spatial_Model.git.

**Funding:** This study was supported in the form of funding by the German Academic Exchange (Project-ID 57417782, Projekt-ID 57599539) awarded to KAS, Sächsisches Staatsministerium für Wissenschaft und Kunst (Project numbers 100257255 and 100613388) awarded to KAS, the Federal Ministry of Education and Research (BMBF) and the DLR (Project-ID 01DQ20002) awarded to KAS. The funders played no role in the design of the study. There was no additional external funding received for this study.

**Competing interests:** The authors have declared that no competing interests exist.

epidemic under different interventions were predicted. In particular, without tightening contact reductions, a strong epidemic wave is predicted during December 2021 and January 2022. Provided the dynamics of the epidemic in Germany, in late 2021 administration of full-dose vaccination to all eligible individuals (e.g. by mandatory vaccination) would be too late to have a strong effect on reducing the number of infections in the fourth wave in Germany. However, it would reduce mortality. An emergency brake, i.e., an incidence-based stepwise lockdown, would be efficient to reduce the number of infections and mortality. Furthermore, to specifically account for mobility between regions, the model was applied to two German provinces of particular interest: Saxony, which currently has the lowest vaccine rollout in Germany and high incidence, and Schleswig-Holstein, which has high vaccine rollout and low incidence.

## Conclusions

A highly sophisticated and flexible but easy-to-parameterize model for the ongoing COVID-19 pandemic is introduced. The model is capable of providing useful predictions for the COVID-19 pandemic, and hence provides a relevant tool for epidemic decision-making. The model can be adjusted to any country, and the predictions can be used to derive the demand for hospital or ICU capacities.

## Introduction

To reduce the impact and burden of the ongoing COVID-19 pandemic and potentially bring a quick end to the worldwide crisis, vaccines were developed at record speed around the globe. Within less than two years, 23 vaccines against COVID-19 received emergency-use approval [1] and several vaccines successfully sought approval for administration among teenagers and children [2, 3]. Among these vaccines, two mRNA-based vaccines, i.e., BNT162b2 and mRNA-1273 received approval for emergency or full use and were widely deployed [4, 5] (see [6] for an overview of the different types of vaccines). Initial hopes were that herd immunity could be reached by mass-vaccination [7, 8]. According to classical SIR-type models, 1 minus 1 by the basic reproduction number ($1-1/R_0$) of the population needs to be immunized to reach herd immunity [9, 10]. However, it is not being thoroughly assessed to which extent the various vaccines protect from infection and transmission [11]. Empirical evidence suggests that vaccinated individuals are less likely to be infected, and less likely to transmit the virus [12, 13]. Moreover, it is unclear how long immunization lasts, and how well vaccines protect against more infectious (larger $R_0$) and aggressive (higher case fatality) SARS-CoV-2 variants that have emerged, spread, and hence increased the herd immunity threshold [7].

Nevertheless, clinical studies proved the effectiveness of vaccines to protect from severe forms of the disease and hence mortality [14, 15]. Therefore, vaccination campaigns remained ambitious. Initially, the focus was on risk groups, when supply of approved vaccines was limited. However, vaccination rates substantially increased with more vaccines having been approved, but vaccination coverage eventually saturated [16, 17] due to widespread vaccination hesitancy, particularly in the younger population (cf. [18, 19]). This forced governments to implement incentives to get vaccinated, e.g., access to restaurants, nightclubs, or mass events without being tested for SARS-CoV-2, or monetary incentives such as vouchers or lotteries

[20, 21]. Moreover, compulsory vaccines for certain professions (e.g., healthcare workers, teachers, flight personnel) or the adult populations are currently subject of debate [22].

Importantly, there are substantial differences across countries regarding access to vaccines and vaccination deployment: Israel pursued a strategy that secured early access to the vaccine and could quickly immunize the population to end contact-reducing measures [23, 24], while Australia advocated a zero-COVID-19 strategy by maintaining strict contact reductions and started vaccination campaigns late [25, 26]. However, these were successful to rapidly immunize a large proportion of the population [25]. Moreover, many low-income countries often neither have good access to vaccines nor the medical infrastructure for deployment [27]. However, in these countries, the risk group of elderly citizens is typically small [28]. Also, the propensity to get vaccinated substantially varies across countries depending on the level of education, trust in the available vaccine, size of the risk groups, etc. For instance, Denmark pursued a strategy of not deploying the AZD1222, when the occurrence of venous thrombo-embolism (VTE) after vaccination was reported, thereby increasing confidence in the safety of the recommended vaccines [29, 30]. Within countries, vaccination rates differ between age groups and regions, with the younger population, which is at lower risk of severe COVID-19 infections, being more hesitant [31, 32]. Due to human contact behaviour being age and location dependent, a sufficient age- and location-dependent vaccination coverage level is important for containing the spread of the virus.

Despite different approaches, countries across the world keep experiencing epidemic waves that ultimately need to be managed by contact-reducing interventions, such as mandatory use of facial masks, frequent testing for SARS-CoV-2, cancellation of mass events, curfews, and lockdowns. In addition, the effect of the spread of new viral variants and the effectiveness of vaccine products against these variants are sources of concern [15]. For instance, South Africa returned millions of doses of the AZD1222 vaccine after the vaccine's effectiveness against the Beta variant was questioned [33].

Predictive models can facilitate the exploration of the effect of different vaccination strategies. However, several models are too simplistic to generate realistic dynamics and/or focus on mathematical details of rather limited relevance for the current pandemic, such as stability analyses of disease-free equilibria (e.g. [34]). Here, a complex and flexible predictive model that accounts for age- and location-dependent contact behavior is introduced to study the impact of different epidemic management strategies. The deterministic SEIR-type model modifies and extends those of the pandemic preparedness tool CovidSim 2.0 [35], as well as the vaccination model of [6]. The model accounts for different viral variants and vaccines introduced at different time points. Viral variants are characterized by different contagiousness and aggressiveness. The propensity to get vaccinated, the onset of vaccine deployment, the vaccination rates age, location, and time dependent. Note that the outcome of the vaccination (complete, partial, or failed immunization) is modeled solely as a property of the vaccine. However, the effectiveness of the vaccine concerning the proportion of severe cases and mortality is age and variant specific. Importantly, vaccinated individuals might fail to immunize, immunize only partially or completely against (at least) one viral variant. The age structure of the model allows to incorporate important age-specific differences, e.g., contact behavior, the fraction of severe cases, mortality rates (reflecting co-morbidities), and prioritization of vaccination. Location-dependent contact-reducing interventions and case isolation measures are included in the model. Contact reductions can follow an incidence-based automatism. We exemplify the model by parameterizing it heuristically to reflect the situation in the Federal Republic of Germany from the onset of the epidemic. We then predict the future epidemic under various epidemic-management scenarios. The whole flexibility of the model cannot be demonstrated within a single article. The purpose of the main text is hence to illustrate how well the model

can be adapted to a real-world situation by choosing intuitive parameters. These parameters are used as default values in the model implementation available at GitHub https://github.com/Maths-against-Malaria/COVID19_Spatial_Model.git. Thus, the text serves as a guidance on how to parameterize the model. Only a verbal description of the model is presented in the main text. A concise mathematical description is presented in S2 Appendix.

## Methods

The spread of different COVID-19 variants is modeled in an age-structured population by an extended SEIR model, based on the pandemic preparedness tool CovidSim (cf. [35]). The model is designed to yield realistic disease dynamics and considers several control interventions, such as general contact reductions, case-isolation, immunization by different vaccines, etc. Note that, the effect of treatment interventions, e.g., remdesivir, steroid- or antibody-treatments, on the disease dynamics is subsumed by the mortality rate. A concise mathematical description is presented in S2 Appendix, in addition to the verbal description provided here.

In the verbal description, we first introduce the different components of the model in subsections. Figs 1 and 2 provide a coarse overview over the model compartments. Readers are advised to first read the verbal description in the main text, before consulting the S2 Appendix. Readers not interested in technical details shall feel free to skip the mathematical description.

### Age and spatial structure

The total population of size $N$ is subdivided into a number of $r$ locations. These can be interpreted as metropolitan and rural areas between which migration is limited, or as different countries (however, this will be difficult to parametrize in general). Furthermore, the population is age-structured, i.e., it is subdivided into $s$ age cohorts in each location.

### Course of the disease

Susceptible individuals are infected by one of finitely many viral variants through contacts with infected individuals in the overall population or from outside the population (external infections). An individual can only be infected by one variant. Not all viral variants are initially present. They are imported from outside the population at some time point (see below and in S2 Appendix).

The course of an infection is qualitatively identical for each age group, location, and viral variant. An infected individual progresses through the (i) latent (not yet infectious, no symptoms), (ii) prodromal (not yet infectious to the fullest extent, no symptoms), (iii) fully-infectious (infectious to the fullest extent, symptoms might start), and (iv) late-infectious phases (Fig 1). Note, in the late infectious phase symptoms will vanish if individuals recover, but intensify if the infections are lethal.

In the fully- and late-infectious phases individuals can either be asymptomatic or symptomatic. A fraction of infected individuals becomes symptomatic (mild or severe symptoms) at the beginning of the fully-infectious phase, whereas the remaining fraction remains asymptomatic. This fraction is dependent on age and the infecting variant. A fraction of symptomatic individuals dies after the end of the late-infectious phase (this fraction again depends on age and variant), while all others (symptomatic or asymptomatic) recover and remain permanently immune against all COVID-19 variants. The average duration of the various phases of the infection are dependent on age and the infecting viral variant.

Implicitly, SEIR models assume exponential-distributed waiting times to progress through the infectious phases. This yields unrealistic epidemic dynamics [6]. To avoid this behavior, each phase of the infection is modeled by a number of equivalent substages through which

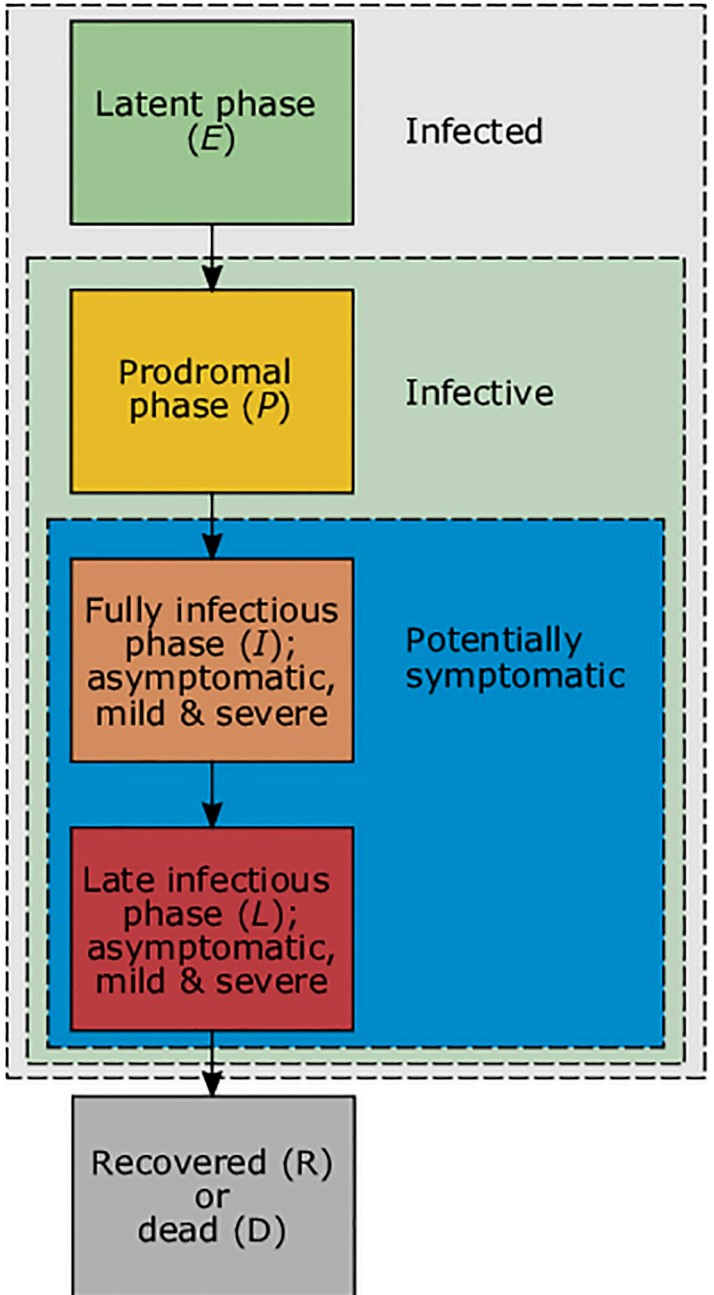

**Fig 1. Progression of COVID-19 infections.** Illustrated are the different phases of COVID-19 infections which finally result in recovery or death.

individuals progress successively. As a consequence, the waiting times to progress between compartments are Erlang-distributed (cf. [6, 35–37]). The number of these substages depends on the phase of the infection, the age group of the infected individuals, and the infecting viral variant.

## Susceptible individuals

Susceptible individuals are further subdivided (see Fig 2) into those that are: (i) not immunized after vaccination and cannot get vaccinated (due to refusal, contraindications, lack of access)

**Fig 2. Subdivision of susceptible individuals.** Susceptible individuals are subdivided by their vaccination status. Each 'class' of susceptible individuals is further stratified by age and location (not illustrated).

(NI); (ii) waiting to be vaccinated (U); (iii) already vaccinated (V), but the outcome of the vaccination is still pending; (iv) vaccinated, but only partially-immune (PI); (v) successfully immunized (Im) (against at least one viral variant). Immune waning is ignored, as vaccinated individuals are assumed to have refreshed their vaccination status.

### Effect of vaccination

Individuals are only vaccinated with one of several vaccines and only one time (subsuming all necessary doses with the respective vaccine product). The rate at which individuals are vaccinated with the different vaccines is age and location dependent, and change over time. This reflects different prioritization for vaccination and supply of the vaccines.

The outcome of the vaccine is not immediate. The waiting time for the vaccine outcome to manifest is dependent on the vaccine and age, reflecting different immune responses among age groups and vaccination schedules among vaccines. The vaccination schedule is subsumed by the waiting time for the vaccination outcome.

After the outcome of the vaccine manifests, a fraction of individuals is (i) successfully immunized (against at least one variant), (ii) partially-immune, or (iii) failed to immunize. These fractions depend on age and the vaccine product, reflecting different effectiveness between the alternative vaccines and among age groups. Susceptibility to the virus is reduced for partially-immune individuals (depending on age group, variant, and vaccine product). Importantly, the effect of immunization against viral variants is vaccine dependent. In particular, individuals can be completely immunized by a vaccine against a specific viral variant, but fail to be

immunized completely against some other variants—however, they will also have reduced susceptibility against these variants. Therefore, in the model all infectious phases of vaccinated individuals are subdivided into the deployed vaccine products and infecting viral variant The probability of having a severe infection is subsumed by the fraction of symptomatic infections, the fraction of symptomatic infections being isolated and mortality in each age group.

If partially-immune individuals become infected, the proportions of symptomatic and lethal infections decrease. This is dependent on the combination of vaccine product and infecting viral variant. Fully immunized individuals might also be infected by certain variants, against which the vaccine is not fully protecting. Then they are treated as partially-immune infected individuals.

Individuals can get infected during the time when the outcome of the vaccine is still pending. The course of the disease is identical to that of non-immunized individuals until the vaccine manifests its effect. In this case, those individuals either recover fast (i.e., they are moved immediately to recovered individuals), get partially-immune and are treated like partially-immune infected individuals, or failed to immunize and are treated like unvaccinated infected individuals.

Individuals waiting to get vaccinated can also be vaccinated when already infected. The model assumes the vaccine has no impact on infected individuals in the fully- and late-infectious phases. Hence, it is ignored that these individuals get vaccinated. However, infected individuals can get vaccinated during the latent and prodromal phases. In these cases, they are treated like vaccinated infected individuals for whom the outcome of the vaccine is still pending.

## Contact behavior and transmission

Susceptibles acquire infection by contacts with infected individuals. The probability to transmit the disease at an encounter depends on the phase of the infection, the viral variant, and the level of (partial) immunization (partially-immune are less likely to get infected than non-immunized susceptibles). Additionally, partially-immune infected individuals are less likely to spread the virus—the reduction in transmissibility depends on the interaction of viral variant and the deployed vaccine. Notably, some infected individuals cannot transmit the virus because they are in quarantine (see Case isolation). Moreover, contacts between susceptibles and infected are not random. Contacts between individuals are assumed to be age- and location-dependent (see S1 Table). In particular, a symmetric matrix between age strata and locations describing the contact behavior has to be specified. The contact matrix changes due to general contact reductions and case isolation (see S4 Table). In the particular model implementation, we use the contact matrices from [38], which provide estimates for contact behavior for over 150 countries, stratifies by contacts at home, work, in school, and at other places. It is not necessary to use these estimates, but they offer a pragmatic choice which allows to model contact reductions in an intuitive way.

Each viral variant $m$ has a particular basic reproduction number $R_0^{(m)}$. All basic reproduction numbers fluctuate seasonally by the same percentage around their seasonal average basic reproduction numbers, and attain their seasonal maximum at the same time. The seasonality depends on the climatic properties of the country the model is applied to and subsumes, e.g., changes in contact behavior due to weather seasons (see Weather adjustment in S2 Appendix for more discussion.

## Case isolation

Only a fraction of symptomatic infections seek medical help. These infections are isolated in quarantine wards, until the maximum capacity is reached, and at home otherwise. Unlike

quarantine wards, home isolation eliminates only a fraction of infectious contacts. Case isolation is maintained only during a specified time interval. In particular, it does not start immediately at the onset of the epidemic (and can be terminated at any time point).

### General contact reduction

General contact reductions reflect curfews, physical distancing, cancellation of mass events, etc. These are modeled by changes in the contact matrix (see Contact behavior and transmission) during certain time intervals. These changes occur during fixed time points or incidence-based. In the latter case, the contact matrix changes in discrete steps if the 7-day incidence surpasses specific threshold values.

### Model implementation

The model as described in detail in S2 Appendix and was implemented in Julia 1.6.3. We used the function *solve* using an adaptive-order adaptive-time Adams Moulton method (*VCABM*) available in the package *DifferentialEquations* [39]. Graphical outputs were created in R [40]. The Julia code of the model implementation with the parameters used in the numerical examples is available at GitHub (https://github.com/Maths-against-Malaria/COVID19_Spatial_ Model.git and https://doi.org/10.5281/zenodo.5718729).

## Results

To illustrate the model, we chose parameters that reflect the Federal Republic of Germany. The parameter choices resemble the epidemic management in Germany until November 2021. To predict the epidemic in the future, different vaccination scenarios and contact reduction, particularly the reintroduction of a COVID-19 emergency brake, were assumed.

### Parameter choices

**Population and contact behavior.** The population of $N = 83787388$ was subdivided into four age strata (see S1 Table) based approximately on census data (in the years 2019–2021) [41]. Regarding the contact behavior, the estimates from [42] were used. (The estimated contact matrices were symmetrized and aggregated to match the age strata used here—exact values are available in the Julia code.) These estimates divided the contact behavior into six categories: (i) contacts at home excluding the oldest age stratum, (ii) contacts at home with the oldest age stratum, (iii) contacts at work, (iv) contacts in school, (v) other contacts excluding the oldest age stratum, (vi) other contacts of the oldest age group. The reason to single out the oldest age group is that contact restrictions can affect them differently, e.g. if elderly citizens depend on nursing or live in closed facilities. Since the estimates for contact behavior are available only for the whole country and estimates for mobility were unavailable to us, the model was not stratified by different locations, i.e., $r = 1$. (In S1 Appendix the effect of more than one location is illustrated by considering only two provinces of Germany, Saxony and Schleswig-Holstein, assuming hypothetical contacts between locations.) Contact reductions were modeled by reducing the contacts at home, work, school, and other contacts during specific time intervals by percentages that roughly reflect the contact reductions imposed in Germany (see S2, S3 Tables). From April 25, 2021 (day $t = 425$) to July 4, 2021 (day $t = 495$) contact reductions were incidence-based, reflecting the COVID-19 emergency brake in Germany [43] (see S4 Table).

**Viral variants.** The wildtype, the Alpha variant (B.1.1.7), and the Delta variant (B.1.617.2) were considered. The wildtype appeared at day $t = 0$ corresponding to February 25, 2020 and has a yearly-average reproduction number $\bar{R}_0^{(\mathrm{WT})} = 3.0$. For the Alpha variant, which was

introduced by external infections on November 21, 2020 (day $t = 290$), a 30% higher average reproduction number $\bar{R}_0^{(A)} = 3.9$ was assumed according to estimates [44]. The average reproduction number of the Delta variant was 40% higher than that of the Alpha variant (cf. [45]), i.e., $\bar{R}_0^{(\Delta)} = 5.46$. The Delta-variant was introduced on June 14, 2021 by external infections (day $t = 475$). The basic reproduction numbers are supposed to fluctuate seasonally 43% ([46]; cf. also [36] for more discussion and references) and peak at day $t_{R_{0_{\max}}} = 300$ (December 21, 2020).

**Seasonal fluctuations in transmissibility.** April 2021 in Germany was the coldest recorded in 40 years, with the average temperature being almost 3˚C colder than the average in the last 30 years [47, 48]. Also, August 2021 was unusually cold with 30% more rainfalls and 30% less sunshine [49]. In the simulations, we adjusted for the unusually cold March/April and August 2021 (see Weather adjustment in S2 Appendix). Namely, seasonal fluctuations in transmissibility are due to factors which correlate with weather (e.g. temperature, UV radiation, amount of outdoor contacts, indoor ventilation) and effect human contact behavior and inactivation of the virus. Weather adjustments corresponds to shifts in the calendar month.

**Vaccines.** The vaccines of BioNTech/Pfizer and Moderna (treated synonymously), AstraZeneca, and Johnson & Johnson were considered. Vaccination with Pfizer-BioNTech started on December 31, 2020 (day $t = 310$) for the oldest age group. AstraZeneca was used from day $t = 360$ (February 19, 2021) and Johnson & Johnson from day $t = 450$ (May 20, 2021). The rates at which the age groups were vaccinated were updated over the course of the simulations (S5 Table).

**Course of infection.** The average durations for the latent, prodromal, fully-infectious and late-infectious phases were equal for all age groups and variants. They were set to $D_E = 3.5$, $D_P = 1$, $D_I = 5$, $D_L = 5$ days, respectively. During the prodromal and late-infectious phases all viral variants were assumed to be transmitted half as efficiently as in the fully-infectious phase ($c_P = c_L = 0.5$, $c_I = 1$; cf. S6 Table).

Individuals partially (or fully) immunized with vaccine $v$ transmit viral variant $m$ less efficiently than unimmunized individuals, i.e., the transmissibility is reduced by a fraction $p_P^{(m,v)}$, $p_I^{(m,v)}$, $p_L^{(m,v)}$, according to S7 Table. If individuals are partially or fully immunized with vaccine $v$, they are less susceptible to variant $m$. In particular, the susceptibility is only a fraction ($g(m, v)$ or $h(m, v)$) of that of unimmunized ones (see S7 Table for the parameter choices).

The percentage of infections that become symptomatic ($f_{\text{Sick}}^{(a,m)}$, $f_{\text{Sick}}^{(\text{PI},a,m,v)}$) depends on age, the status of immunization (U, V, Im, PI, NI), the vaccine that yielded immunization and the infecting viral variant (see S8 Table for parameter choices).

**Reported quantities.** We report disease incidence and prevalence. Disease prevalence is defined as the sum of all infected individuals (in the latent, prodromal, fully-infectious, and late-infectious phases). By incidence, we refer to 7-day incidence, which is the number of new cases within the last 7 days per 100 000 individuals (see definition in S2 Appendix).

## Dynamics

The model dynamics quantitatively mimic the observed dynamics in Germany [50]. The model accurately reflects the first, second, and third COVID-19 waves in spring 2020, fall 2020, and spring 2021, respectively in terms of active infections (Fig 3A and 3B—stratified by age and viral variants, respectively), as well as mortality (Fig 3C) and the vaccination progress (Fig 3D). Incidence in the model outcome subsumes unreported cases. Therefore, it exceeds the empirically observed incidence. The fraction of unreported cases depends on testing capacities, the propensity to get tested, and testing policies, which change over time.

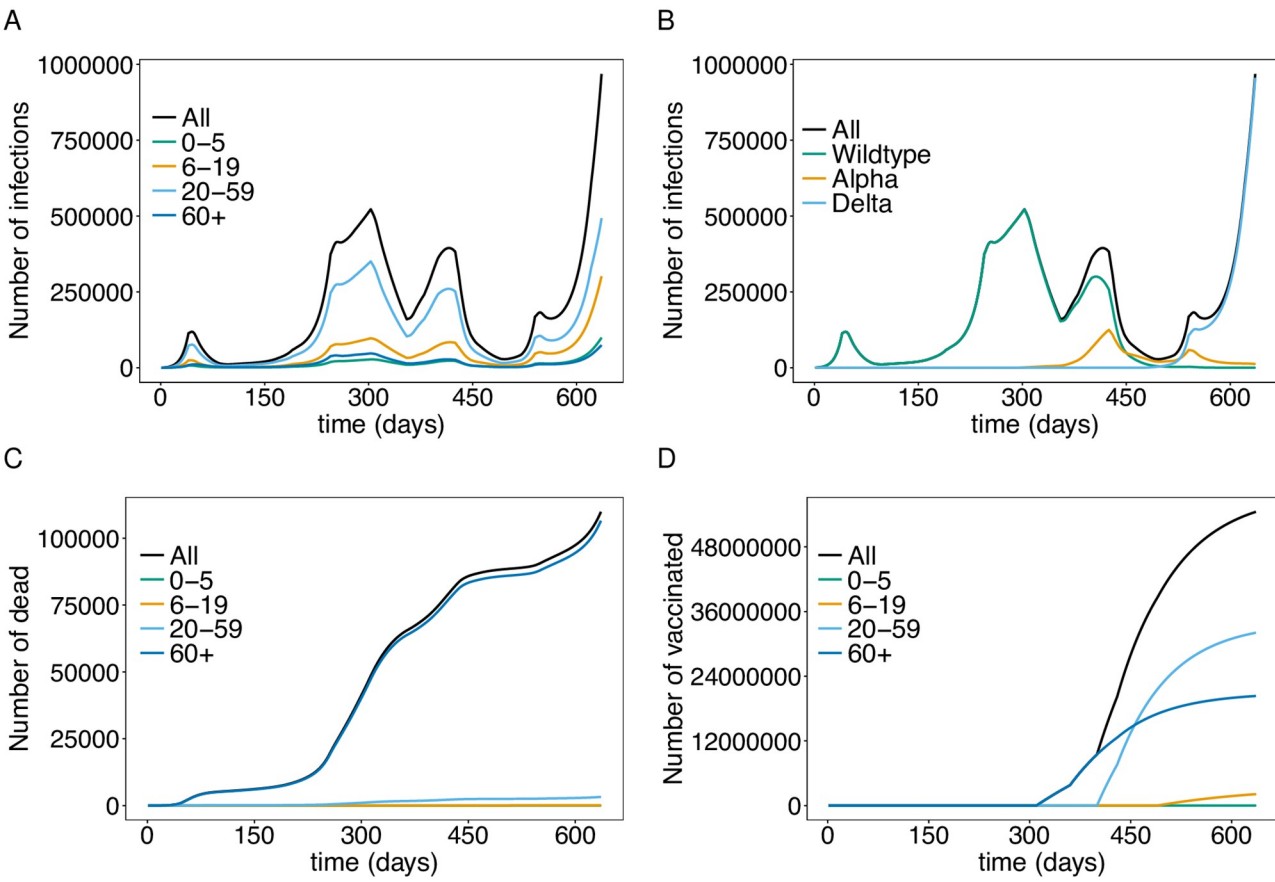

**Fig 3. Epidemic dynamics in Germany until fall 2021.** The total numbers of infected (latent ($E$), prodromal ($P$), fully- ($I$), and late-infectious ($L$) individuals) stratified by age group (A) or viral variants (B). (C) The total numbers of dead individuals ($D$) stratified by age group. (D) Vaccination progress by age group. The parameters used for the simulations are listed in S1 to S11 Tables. Simulation time is from day $t = 0$ to day $t = 636$.

The 'first wave' in Germany lasted from mid-March 2020 to mid-May 2020, had a peak of approximately 100 000 active infections and was controlled by a hard lockdown. The 'second wave' started in September 2020 when the basic reproduction number increased due to seasonal fluctuations. The increase in incidence was reduced as stricter contact-reducing measures were introduced in early November 2020 (day $t = 250$). The second wave peaked around the end of the year 2020, during the Christmas holidays (day $t = 303$), followed by a decline because of stricter contact reductions. However, in early March 2021 (day $t = 370$) incidence increased again as the Alpha variant became predominant (Fig 3B), launching the 'third wave', which started to decline in late April 2021 (day $t = 425$), when an incidence-based federal emergency brake was introduced. In June 2021 the Delta variant became predominant (Fig 3B) and a fourth COVID-wave started in late August, which was unusually cold (see Contact behavior and transmission in Methods). A slight decline in incidence followed after weather normalized, and contact reductions were implemented. Finally, incidence increased again as of mid-October 2021 (day $t = 597$) due to seasonality in transmission.

Importantly, the fraction of infections attributed to children and teenagers is higher in the third and fourth wave than in the first and second wave. In the fourth wave, the number of infections among young children (<5 years) is higher than that among elderly (Fig 3A). This is because schools were open during the later waves, and partly due to the vaccination coverage among the older population [51, 52]. This trend can reverse, depending on the epidemic

management strategy (see below). Note that in practice, the number of unreported cases will be higher in young children than adults.

Vaccination campaigns were launched in late December 2020 with the vaccine of BioN-Tech/Pfizer, prioritizing senior citizens [53]. The vaccination rates increased successively, as additional vaccines were approved [54]. Large-scale vaccination campaigns started in mid-May 2021 for all adults. In June 2021, the vaccine from Janssen (Johnson & Johnson) became available, and vaccines received approval for teenagers. Over the summer of 2021 the speed of vaccination campaigns started to stagnate, and reached a level of approximately 66% of the population being fully vaccinated (Fig 3D).

**Predictions for the flu season 2021 and epidemic management.** The course of the epidemic depends crucially on the epidemic management during the 2021 flu season. Different scenarios were compared here: (i) a continuation of the interventions of November 2021 with no further implementations of contact reducing measures; (ii) a return to the emergency brake from April 2021 after November 22, 2021; (iii) the counterfactual scenario in which the emergency brake from April 2021 would have been continued; and (iv) the continuation of the current (end of November 2021) measures with additional schooling from home, i.e., schools remained closed.

As of November 2021, a strong fourth epidemic wave is predicted that will last until May 2022 if no further contact-reducing measures are implemented in fall/winter 2021. Incidence is predicted to be almost seven times higher (7-day incidence of 1 900 including unreported cases) than in the second wave, with an epidemic peak around early January 2022 (Figs 4A, 4C and 5A).

Returning to the emergency brake of April 2021 would lead to a strong and swift decrease in the number of cases. The epidemic peak would reach a level about twice that of the second wave (a 7-day incidence of approximately 600 including unreported cases). Numbers would steadily decrease until summer 2022 (Figs 4A and 5B). This yields a significant reduction in mortality (Fig 4B). The strongest effect would have occurred if the emergency brake from April 2021 would have never been exempt (Figs 4A, 4B and 5B). A fourth wave would also have been unavoidable under this scenario. This is due to seasonal fluctuations in transmissibility and the incidence-based contact reductions. Importantly, the mortality would stagnate at a level, that was already reached in fall 2021 (Fig 4B).

Schooling from home, i.e., closing schools, would also have an immediate effect, which however is less effective than the emergency brake. It mainly slows down the increase in incidence, which leads to a reduction of the epidemic peak of approximately 50% (Figs 4C and 5C). However, mortality is not efficiently reduced, leading to a doubling of the current level (Fig 4D).

**Effect of the mutants.** During initial vaccine development and phase III clinical studies in 2020, the wildtype was still the dominating viral variant. The epidemiological strategy was to reach herd immunity by large-scale vaccination campaigns to end the epidemic. However, when the first vaccines were approved, more aggressive viral variants started to spread. Fig 6 shows the course of the epidemic if the mutations had not occurred, assuming a continuation of interventions from early November 2021 (scenario i). If none of the mutations had appeared, the third wave would have been milder and the fourth wave would not have occurred. More precisely, the strict contact reductions after the holiday season 2020 would have been sufficient. Furthermore, the vaccination campaigns (Fig 6A) would have been sufficient to avoid a fourth wave and mortality would have stayed below, 100 000 (Fig 6B) under the current contact reductions, if the Delta -variant had not occurred.

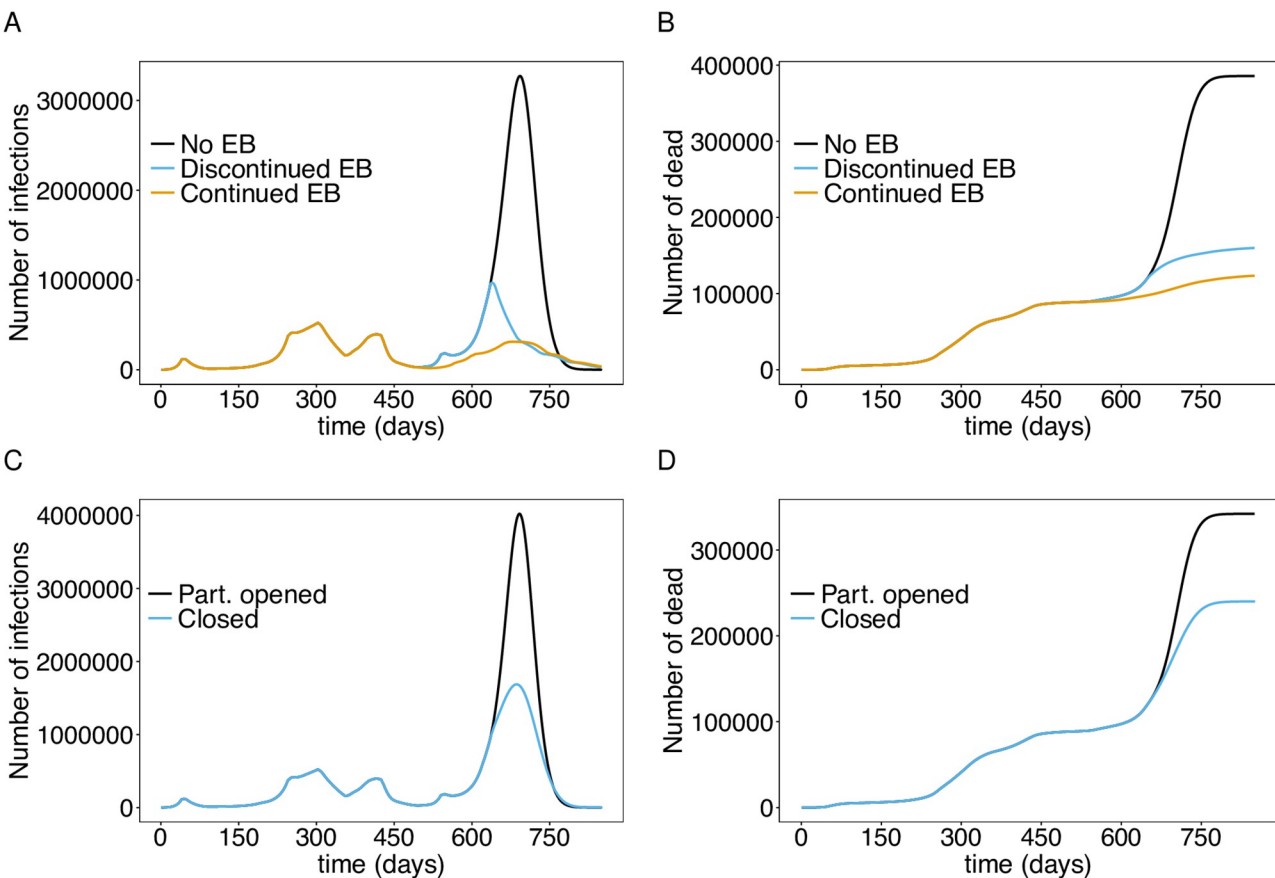

**Fig 4. Impact of emergency brake (EB) and schooling from home.** Shown are the total numbers of infected ($E$, $P$, $I$, $L$) individuals (A) and deaths (B) assuming a continuation of the current interventions, a second emergency brake starting Nov. 22 which is continued to the end of the simulation, and the hypothetical effect of a continuation of the emergency brake implemented in April 2021. (C) shows the total number of infections under the continuation of the current interventions and additional schooling from home (closing schools) until the end of the simulation runs and the corresponding cumulative number of deaths (D). The parameters used for the simulations are listed in S1 to S11 Tables.

Notably, herd immunity would have been reached if the Delta -variant had not emerged (Fig 6A), and mortality would have stagnated at the level after the third wave at below 100 000 (Fig 6B).

**Mandatory vaccine.** We quantified the effect of mandatory vaccination of individuals older than 12 years of age starting from November 22, 2021 ($t = 636$). This scenario assumes that the vaccination rates are increased substantially after $t = 636$ to reach 95% of the population older than 12 years within a relative short time period (cf. S5 Table). Mandatory vaccination leads to a noticeable reduction in the number of infections (Fig 7A), however this intervention (at this point in the epidemic) is much less effective than school closures or an emergency brake (which, however, have substantial economic costs). The reason is that immunization is not immediate and thus not very effective during an ongoing epidemic wave. However, it leads to a noticeable reduction in mortality (Fig 7B).

**Weather adjustments.** To quantify the effect of the extremely cold months in 2021, we performed simulations without the weather adjustments, assuming no additional contact reductions after November 22, 2021 (see above scenario i). Importantly, the third wave in April 2021 would have been substantially less pronounced (Fig 8A). The occurrence of the third wave due to the spread of the Alpha variant would not have been avoided. However,

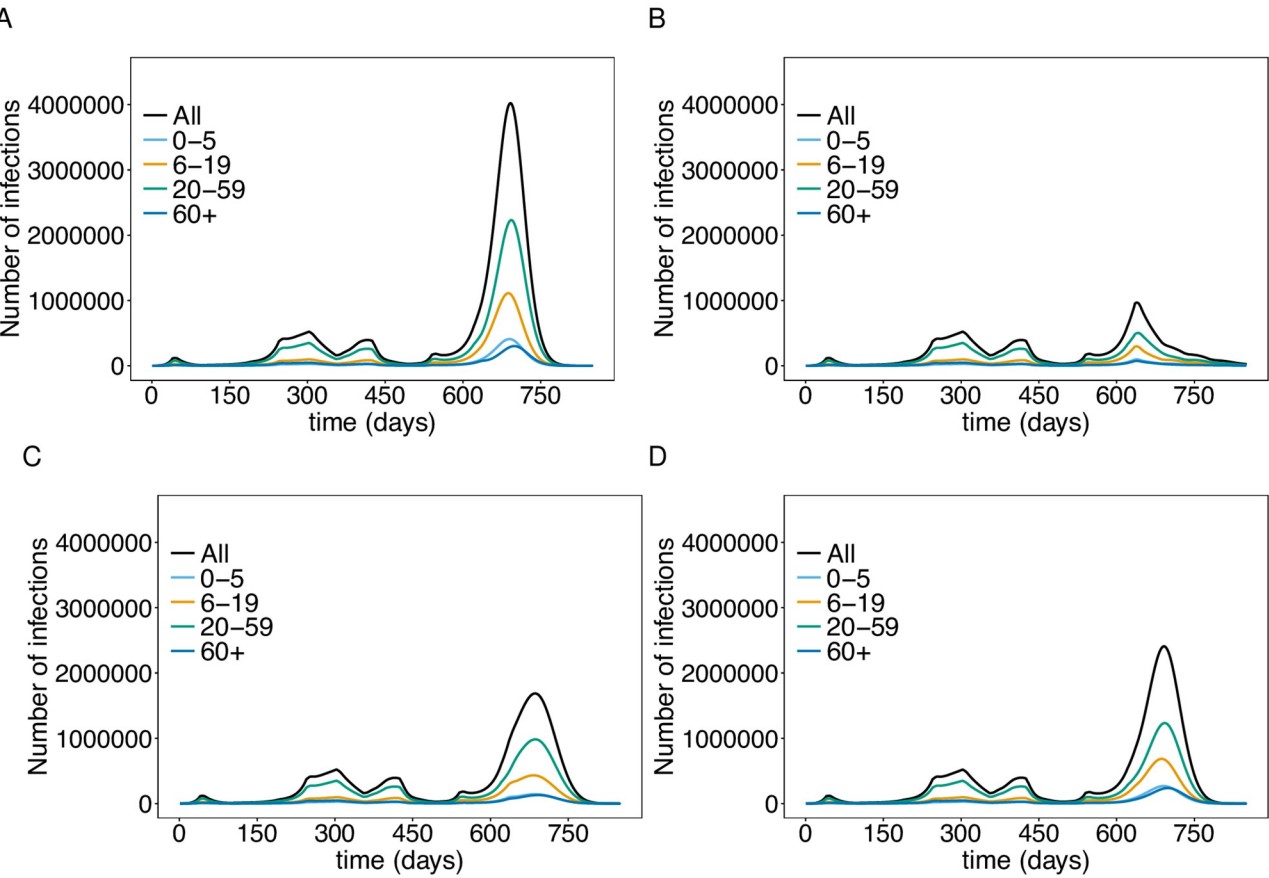

**Fig 5. Number of infections stratified by age.** The total number of infections and the number of infections per age group assuming continuation of the interventions from November 22, 2021 (A), an emergency brake (B), schooling from home (C), and mandatory vaccination for teenagers and adults (D). The parameters used for the simulations are listed in S1 to S11 Tables.

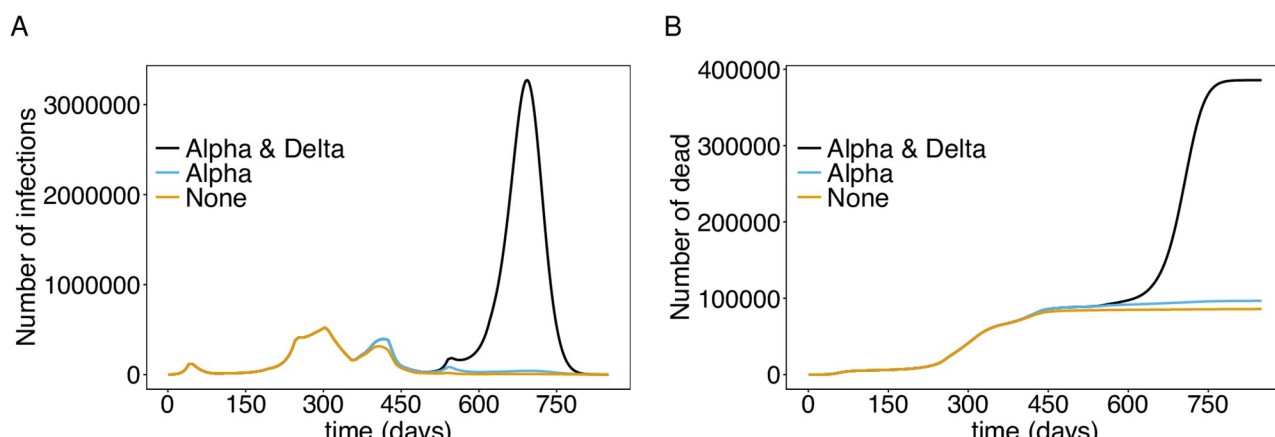

**Fig 6. Role of viral variants.** (A) Shown are the total numbers of infected individuals under a continuation of the interventions from November 22, 2021, assuming none of the viral variants, only the Alpha-variant, or the Alpha- and Delta-variant would have occurred. (B) Shown are the corresponding number of deaths. The parameters used for the simulations are listed in S1 to S11 Tables.

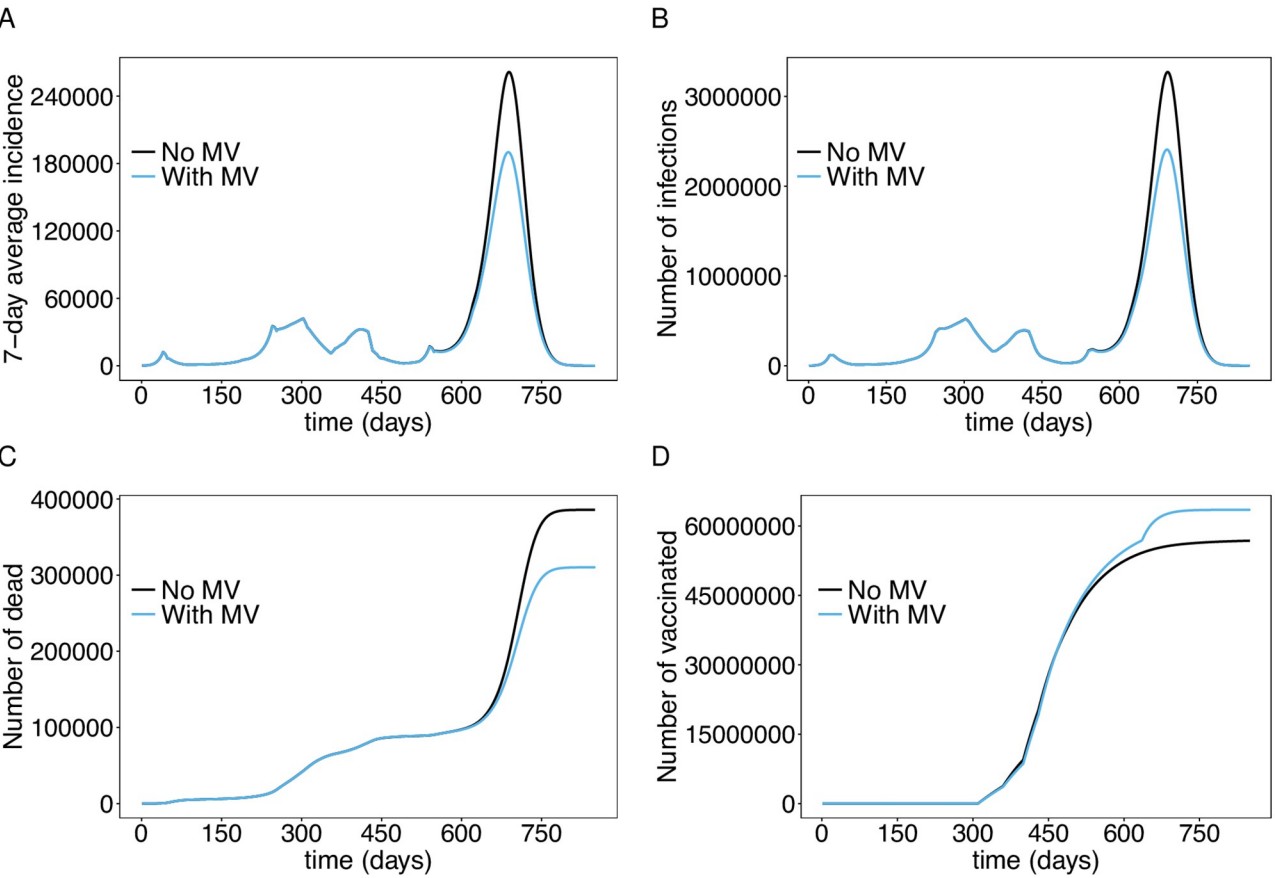

**Fig 7. Effect of mandatory vaccination (MV).** Shown are the 7-day average incidence (A) and total numbers of infected individuals (B) with and without mandatory vaccination of individuals older than 12 years of age after November 22, 2021, assuming a continuation of the interventions of November 2021. Further shown are the corresponding numbers of deaths (C) and vaccinated individuals (D). The parameters used for the simulations are listed in S1 to S11 Tables.

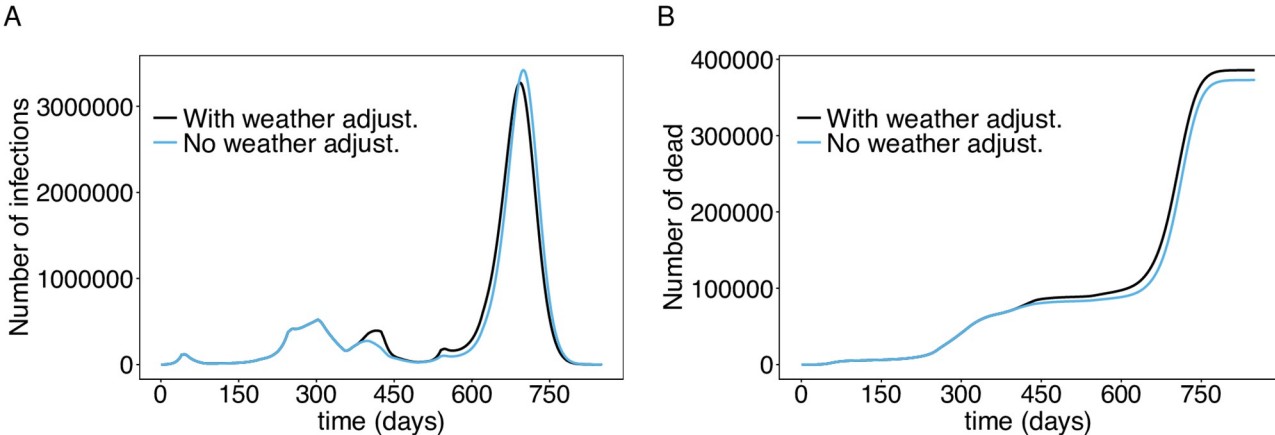

**Fig 8. Effect of weather adjustments.** (A) Shown are the total numbers of infected individuals assuming a continuation of the current contact reductions (scenario i), with and without the weather adjustments in 2021. (B) Shown are the corresponding deaths. The parameters used for the simulations are listed in S1 to S11 Tables.

mortality would have been lower (Fig 8B). If August 2021 had not been as cold (no weather adjustment) the onset of the fourth wave would have been slightly reduced and delayed. The scenario simulated initially yields a slower increase in incidence, followed by a slightly higher and later epidemic peak (due to the seasonal increase in the basic reproduction number). However, mortality would have stayed lower (Fig 8B).

## Discussion

By November 2021, over 20 COVID-19 vaccines were approved and large parts of the population in industrial nations were vaccinated, and governments were eager to relax contact-reducing interventions to return to normalcy. For instance, Denmark lifted all COVID-19 restrictions in summer 2021 when over 75% of the population was vaccinated [55, 56]. However, the Danish government had to reinstate restrictions as COVID-19-related hospitalizations started to rise again [57]. Similarly, after Israel 're-opened the country' in summer 2021, it experienced a strong fourth COVID-19 wave, although a large part of the population was already vaccinated [58–60]. The progress of vaccination campaigns differs substantially across the globe. It is not everywhere as fast as in Israel or Denmark, e.g., in many countries in sub-Saharan Africa, relatively few people are vaccinated [61]. This is partly due to the demography, with 50% of the population being younger than 18 years of age and less than 2% of the population being older than 70 years of age [62, 63]. Hence, the COVID-19 risk group is small, while vaccines are not even approved for the majority of the population. This leads to a different assessment of the benefits of vaccines. In any case, the effects of vaccination campaigns in different countries across the globe are uncertain and predictive models are an important tool to assist epidemic decision-making.

Here, a flexible predictive model for the COVID-19 epidemic was introduced to predict the epidemic dynamics under various vaccination and disease-management strategies. The model is a sophisticated extension of an SEIR-type model, in which an arbitrary number of different viral variants can be modeled. These differ not only in their basic reproduction number, but are allowed to differ in the course of the disease (i.e., in terms of mean and variance of the durations of the latent, prodromal, fully-, and late-infectious phases), the infectiveness in the different disease phases, and the likelihood of causing severe infections. Even if these differences are/were not properly supported by empirical evidence, this flexibility is necessary to make the model applicable to mutant variants that might occur in the future and potentially differ in the aspects mentioned. In the numerical examples only the viral variants relevant in Germany until November 2021 were considered, i.e., the wildtype, Alpha variant (B.1.1.7), and Delta variant (B.1.617.2). New mutants enter the model by an external force of infection, i.e., infective contacts with individuals from outside the considered population. Importantly, the model is age-stratified, to account appropriately for the age-dependent contact behavior, which substantially influences the dynamics of the epidemic [38]. The age structure of the population also needs to be addressed to adequately model vaccination campaigns. In fact, many countries prioritize certain age groups in dispersing the vaccine, and most of the vaccines are not approved or recommended for all age groups (cf. [15–17]). The model further accounts for time-dependent vaccine supply. Moreover, the model captures the fact that vaccine products differ in their effectiveness (in terms of protection from severe disease), vaccination schedule, and the protectiveness from infection and transmission. Namely, it is empirically proven that vaccine products have different effectiveness against each viral variant (cf. [15]) and COVID-19 vaccines do not fully protect from infection and transmission [64]. A potential reason could be that vaccines trigger the production of immunoglobulin G, while immunoglobulin A is the antibody class being prevalent in the mucous membrane, through which the virus enters the

human body [65, 66]. The model allows specifying how much each vaccine protects against the various viral variants regarding infectiveness, transmissibility, occurrence of symptoms, and mortality. The model is also spatially stratified. Hence, it can be used as a pandemic model subsuming several countries. Notably, the spatial stratification can also be important when the model is applied to a single country. For instance, many countries in sub-Saharan Africa are large by area, but the inhabited parts are often concentrated on a handful of metropolitan areas. In such countries, the contact behavior is not random, neither between age groups nor between locations (metropolitan areas). Even in countries, in which the population distributes evenly across the area, contacts between individuals might depend sensitively on mobility. Such a spatial structure can often be modeled on a continuous scale (mathematically in terms of partial differential equations—PDE). However, the situation is also captured by the model introduced here, because it can be interpreted as a discretization of a PDE model.

A model as flexible as the one presented here involves numerous parameters. However, the proposed model is still easy to parameterize, as demonstrated by the numerical examples. Estimates for many parameters, such as the basic reproduction numbers of the different viral variants or the duration of the different phases of the infection, exist. While the model allows many parameters to be dependent on age, location, viral variant, or a specific vaccine product, these do not necessarily need to be chosen differently. E.g., if no empirical evidence exists for the disease progression to depend on age and the viral variant, the respective parameters can be chosen to be identical for all age groups and viral variants (as done in the examples), which substantially reduces the number of model parameters. Typically, if parameters are clearly age-dependent, e.g., the likelihood of severe or lethal infections, age-stratified estimates exist in the literature. Other parameters, e.g., the onset of vaccination campaigns or the vaccination rates per age group and location, can be easily estimated from official data.

The most difficult parameters involved are those defining the age- and location-dependent contact behavior. For the age-dependent contact behavior, estimates for 152 countries are given in [42]. The contacts are subdivided into interactions happening at home, at work, in school, and at other places. These estimates can be further aggregated or disaggregated as needed. However, spatial contact behavior is more difficult. In general, mobility can be estimated from cell-phone data or air-traffic data [67, 68]. While this might be simple for countries in which the population is concentrated in a few metropolitan areas, in more evenly distributed populations it will be much more difficult to estimate contacts at a fine-grained level.

As an example for the model's applicability, we parameterized the model to reflect the situation in Germany. Because adequately estimating mobility data would have exceeded the scope of this article, we modeled Germany as a single location. The contact behavior was estimated from the contact matrices in [42], which allowed us to model the effects of specific contact reductions such as school closures, working from home, etc.

The numerical results for Germany accurately reflect the past dynamics of the epidemic and a severe 'fourth wave' is predicted. According to the model, the introduction of mandatory vaccines for individuals older than 12 years of age at the current state of the epidemic in Germany would not have a substantial effect on the number of infections. However, it has a substantial effect on reducing mortality. Closing schools would lead to a stronger reduction in the number of infections, but does not lead to an equally strong reduction in mortality. Not surprisingly, an incidence-based emergency brake, i.e., a stepwise lockdown based on incidence thresholds, would be the most efficient measure to reduce incidence and mortality. The epidemic waves in Germany are due to seasonal fluctuations in transmissibility. More precisely, the basic reproduction numbers of the viral variants fluctuate seasonally around a yearly average. Importantly, the weather in Germany in 2021 was characterized by two cold weather

extremes, which are accounted for in the simulations. We also demonstrated the effect of these 'weather adjustments'.

To demonstrate the applicability of the model to multiple locations, in S1 Appendix we presented predictions for two spatially separate German provinces, i.e., Saxony and Schleswig-Holstein, assuming hypothetical mobility between these two locations. The two provinces were selected, because (i) they are distant and mobility exchange is sufficiently low to heuristically define the contact behavior between locations, rather than relying on accurate mobility data; and (ii) these provinces differ in a number of important characteristics. In the end of November 2021, Saxony had the lowest vaccination rollout among all German provinces, while Schleswig-Holstein has one of the highest. Moreover, Saxony had the highest COVID-19 incidence in Germany, although the fourth epidemic wave started there later than in other provinces. In Schleswig-Holstein the fourth wave started relatively early, but incidence in late November 2021 was lower than in most other provinces. Infective contacts with other provinces and countries are accounted for by location-specific external forces of infection. The model dynamics accurately reflect the characteristics of the epidemic in both provinces, demonstrating the ability of the model to realistically reflect spatial characteristics of the epidemic.

While the results demonstrate the applicability of the model, predictions can be improved by more accurate parameter estimates, which are beyond the scope of this article. The model is flexible enough to be tailored to any country and sub-region. Possible extensions and generalizations are manifold. From the model's structure, it is straightforward to do these and implement them in the code. Such generalizations include temporary immunity, contact restrictions only for unvaccinated individuals or booster vaccine shots.

## Supporting information

**S1 Appendix. Predictions for Saxony and Schleswig-Holstein modeled together.**
(PDF)

**S2 Appendix. Mathematical description of the model.**
(PDF)

**S1 Table. Age-stratified population size of Germany (GER).**
(PDF)

**S2 Table. Timeline of contact reduction measures chosen for the simulations of Germany.**
(PDF)

**S3 Table. Contact reduction parameters chosen for the simulations.**
(PDF)

**S4 Table. Parameters describing incidence-based contact reductions for emergency brake conditions in Germany.**
(PDF)

**S5 Table. Parameters describing the vaccination rate.**
(PDF)

**S6 Table. Parameters describing disease progression.**
(PDF)

**S7 Table. Parameters describing vaccination outcome and immunity.**
(PDF)

**S8 Table. Parameters describing disease severity and mortality.**
(PDF)

**S9 Table. Variables describing initial values of individuals in non-infected compartments.**
(PDF)

**S10 Table. Variables describing initial values of individuals in infected compartments.**
(PDF)

**S11 Table. Parameters describing external-, seasonal-factors, control measures, and contagiousness.**
(PDF)

**S1 Fig. Flow chart: In the flow chart, Greek letters indicate rates.** Capital letters $S$, $E$, $P$, $I$, $L$, $R$ and $D$ stand for susceptible, latent (early), prodromal, fully-infectious, late-infectious, recovered, and dead, respectively. Overlapping squares indicate equivalent compartments, which are surpassed successively (the number 1 in the sub-script indicates the first of the respective sub-states). The sub-scripts $a$ and $l$ indicate the age group and location. In the super-scripts, $m$ indicates the infecting variant and $v$ the different vaccines. Moreover, in the super-scripts 'U' indicated unvaccinated individuals waiting to be vaccinated, 'V' indicates vaccinated individuals for which the outcome of the vaccine is pending, 'PI' partially-immune individuals, and 'NI' individuals that are unvaccinable or failed to immunize. Finally, 'Inf' indicates that individuals recovered from infection, and 'Im' implies that individuals were completely immunized against at least one variant.
(PDF)

**S2 Fig. Effect of emergency brake—two locations model.**
(PDF)

## Acknowledgments

We want to dedicate this work to all voluntary participants in the COVID-19 vaccination trials. We also want to dedicate it to the victims of the SARS-CoV-2 virus. Our grief is with the friends and families of the dreadful disease. The authors gratefully acknowledge the helpful comments of two anonymous reviewers.

## Author Contributions

**Conceptualization:** Kristan Alexander Schneider.

**Data curation:** Kristan Alexander Schneider, Henri Christian Junior Tsoungui Obama, Nessma Adil Mahmoud Yousif.

**Formal analysis:** Kristan Alexander Schneider, Henri Christian Junior Tsoungui Obama, Nessma Adil Mahmoud Yousif.

**Funding acquisition:** Kristan Alexander Schneider.

**Investigation:** Kristan Alexander Schneider, Henri Christian Junior Tsoungui Obama, Nessma Adil Mahmoud Yousif.

**Methodology:** Kristan Alexander Schneider, Henri Christian Junior Tsoungui Obama, Nessma Adil Mahmoud Yousif.

**Project administration:** Kristan Alexander Schneider.

**Resources:** Kristan Alexander Schneider.

**Software:** Kristan Alexander Schneider, Henri Christian Junior Tsoungui Obama, Nessma Adil Mahmoud Yousif.

**Supervision:** Kristan Alexander Schneider.

**Validation:** Kristan Alexander Schneider.

**Visualization:** Kristan Alexander Schneider, Henri Christian Junior Tsoungui Obama.

**Writing – original draft:** Kristan Alexander Schneider, Henri Christian Junior Tsoungui Obama, Nessma Adil Mahmoud Yousif.

**Writing – review & editing:** Kristan Alexander Schneider, Henri Christian Junior Tsoungui Obama, Nessma Adil Mahmoud Yousif.

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
