## [Decision Letter · Decision Letter 0]

6 Sep 2022

PONE-D-21-37770Predicting the impact of COVID-19 vaccination campaigns - a flexible age-dependent, spatially-stratified predictive model, accounting for multiple viral variants and vaccinesPLOS ONE

Dear Dr. Schneider,

Thank you for submitting your manuscript to PLOS ONE. After careful consideration, we feel that it has merit but does not fully meet PLOS ONE’s publication criteria as it currently stands. Therefore, we invite you to submit a revised version of the manuscript that addresses the points raised during the review process.

We look forward to receiving your revised manuscript.

Kind regards,

Mohammad-Reza Malekpour

Academic Editor

PLOS ONE

Journal Requirements:

“This study was supported in the form of funding by the German Academic Exchange

(Project-ID 57417782, Projekt-ID 57599539) awarded to KAS,

Sächsisches Staatsministerium für Wissenschaft

und Kunst (Project number 100257255) awarded

to KAS, the Federal Ministry of Education and

Research (BMBF) and the DLR (Project-ID

01DQ20002) awarded to KAS.”

Reviewers' comments:

Reviewer's Responses to Questions

**Comments to the Author**

1. Is the manuscript technically sound, and do the data support the conclusions?

Reviewer #1: Yes

Reviewer #2: Partly

2. Has the statistical analysis been performed appropriately and rigorously? 

Reviewer #1: Yes

Reviewer #2: I Don't Know

3. Have the authors made all data underlying the findings in their manuscript fully available?

Reviewer #1: Yes

Reviewer #2: Yes

4. Is the manuscript presented in an intelligible fashion and written in standard English?

Reviewer #1: Yes

Reviewer #2: Yes

5. Review Comments to the Author

Reviewer #1: This is a comprehensive and flexible COVID-19 transmission model with a user front-end and open access code. While the authors state on the last line of the Discussion that temporary immunity can be implemented, the authors are encouraged to state if and when this improvement will be made to capture naturally acquired and vaccine-induced waning immunity. As well, hospital admissions should be reported as part of these findings.

MAJOR COMMENTS

1. Abstract, Methods and findings: the authors are encouraged to report the projections for the reduction in hospital admissions (i.e., health system burden and health system stress) in addition to the effect on infections and mortality.

MINOR COMMENTS

2. Abstract, Methods and findings: the section “and predicted the future dynamics of the epidemic under different interventions” could be revised to “and future dynamics of the epidemic under different interventions were predicted.”

3. Abstract, Methods and findings: the authors should consider indicating when they predict a strong epidemic wave in Germany, i.e., at least in which season of 2022 this wave is predicted.

4. Abstract, Methods and findings: could the authors be more specific about how they define ‘mandatory vaccination’ (i.e., dose number, target group, coverage, and timing).

5. Abstract, Conclusions: the aspect of deriving economic collateral damages is only introduced in the conclusion as a feature of the model, and readers will need to have further elaboration as to what this encompasses. For example, does this tool consider labour force loss, border closure implications on trade, effect on domestic growth, impact on tax revenues, increased expenditures on health and other areas, decline in revenue, etc.?

6. Introduction, line 1: do the authors think it is fair to state the COVID-19 vaccines were developed to bring a ‘quick end’ to the pandemic? Developers and planners were likely well aware that restrictions around supply and demand, rollout, and vaccine efficacy in light of emerging new variants with changing would unfortunately not result in a quick end of a global pandemic. Perhaps vaccines were developed to reduce the impact and burden of the pandemic and potentially towards the eventual end of the pandemic.

7. Introduction, lines 3 and 4: the sentence ‘With BNT162b2 and mRNA-1273, two mRNA-based vaccines received approval for emergency or full use and were widely deployed’ needs to be reworded, e.g., BNT162b2 and mRNA-1273, are two mRNA-based vaccines the received approval for emergency or full use and were widely deployed’. Also, were these vaccines widely deployed evenly across all world continents? In the next sentence, are these 2 vaccines included in the set of 23 vaccines? The authors may consider listing the types of vaccines (mRNA, vector (disabled adenovirus), protein subunit, yeast fermentation). The section ‘for teenagers and children’, should likely read something like ‘for administration among teenagers and children’.

8. Introduction, line 7: the authors may consider reframing their statement that it is ‘unclear to which extent the various vaccines protect from infection and transmission’. While the global clinical community have not been able to track the efficacy of the newly developed COVID-19 vaccines over relatively long periods, as clinical trial evidence has been published, as stated in the next sentence, i.e., empirical evidence suggests, perhaps this could be framed as evidence needs further follow-up or examination or validation. Unclear does not seem to be the appropriate term to me. Duration of protection (i.e., decay) being unclear seems to be a fair assessment.

9. Introduction, line 20: the statement that vaccination coverage eventually saturated is quite broad and unspecific. Is there a certain coverage level(s) and timeline of this saturation?

10. Introduction, lines 42-44: this sentence could be slightly reworded to something like ‘Due to human contact behaviour being age and location dependent, a sufficient age- and location-dependent vaccination coverage level is important for containing the spread of the virus.’

11. Introduction, line 56: specify that this is an individual-based model.

12. Introduction, line 61: the authors may wish to elaborate on the statement that vaccine efficacy is location dependent because of location-specific contact dynamics, and not because efficacy alone differs by location.

13. Methods, line 77: does the model also captured risk by age-structure in the form of co-morbidities?

14. Methods, line 81: add a comma before ‘etc.’ It may be useful to include if the model captures treatment interventions within ‘etc.’, for example antivirals like paxlovid.

15. Methods, line 86: if more than one country is selected, how is migration handled? Can more than one country be selected with further location disaggregation by region with urban and rural specification?

16. Methods, line 93: can the authors describe (if not done so in the text beyond this point) or elude to here that this will be further described, what the profile of emerging new variants that become dominant looks like in a given population/location?

17. Methods, line 98: in the late-infectious period, have symptoms, if any presented, been reduced or is the individual assumed to be asymptomatic during this stage?

18. Methods, line 105: can the authors please provide a rationale for why they chose to assume people who have been infected become permanently immune? This does not reflect reality. Immunity wanes and people can become reinfected.

19. Methods, line 120: following successful immunization, is there immunity waning? If so, what is the profile of the decay (e.g., exponential or biphasic, with a specified half-life, etc.).

20. Methods, line 122: Can the authors state whether all necessary dose with a respective vaccine product represents primary series (i.e., one or two doses) or also booster doses (and if so, how many booster doses – it is only noted on line 482 that booster shots can be implemented, this should be made clear earlier)? Also, since doses are given only one time, that this does not represent the real-life peak and decay of vaccine-induced immunity, and the implications this simplification will have on projections.

21. Methods: As an overall suggestion, reference to relevant sections from the Mathematical Appendix could be referred to appropriately throughout the Methods section.

22. Methods, line 141: are the proportion of those symptomatic further disaggregated into clinical and severe (prior to being lethal)?

23. Methods, line 146: this sentence should be reworded something like ‘Individuals can get infected during the time when the outcome of the vaccine is still pending.’ As written, the comma between the two clauses of the sentence makes interpretation unclear.

24. Methods, line 149: again, is there decay following partial immunity?

25. Methods, line 157: is the period during which that outcome of the vaccine is still pending define by the user, or is this specified in the model similarly for all vaccine products?

26. Methods, lines 161-162: the punctuation needs to be corrected to ‘immunization (partially-immune are less likely to get infected than non-immunized susceptibles).’ the first period was omitted and a close bracket was inserted before the second period.

27. Methods, line 168: can the authors elaborate on how locations are defined, e.g., household, community, school, workplace or other location? Are there probabilities of transmission for each location, it is assumed so, are these listed anywhere? Later in the Results section ‘Population and contact behavior’ the locations are described, this seems to be better suited for description in the Methods section.

28. Methods, line 173: perhaps the authors could add a sentence describing the effect seasonality plays on basic reproduction numbers, for example, during warmer seasons (spring and summer) individuals tend to remain outdoors more, therefore contact numbers are reduced and basic reproduction is decreased by a given percentage, with the opposite assumed for cooler seasons.

29. Methods, line 180: how long after the start of the epidemic is it assumed that case isolation effect will be modelled?

30. Methods, line 182: following convention used by the WHO, social distancing should be referred to as physical distancing.

31. Results, line 203: is this population size representative of the population of Germany for the year 2020 for example? Please clarify.

32. Results, line 213: the authors state that ‘estimates for mobility were unavailable to us’, could not the Google Mobility data (as noted on line 438), which was available prior to November 2021, have been used? This could have been adapted to the contract matrix design. Later on lines 442-442, the authors state that ‘Because adequately estimating mobility data would have, exceeded the scope of this article. This should be stated earlier.

33. Results, line 220: are these ‘t’ values reported in number of days in the burn-in period? Please include units. The same applies for this instance on line 227, and throughout.

34. Results, lines 221-222, S4 table: the title should reflect that these parameters where assumed for Germany, if true. For the asterisk definition of incidence, it would be useful to state if these are (confirmed and unreported) SARS-CoV-2 cases. As well for ‘per 100 000’ note that these are individuals. Is there a reference to support how incidence thresholds were set?

35. Results, line 224: by convention variant names ‘Alpha variant’ are not hyphenated as was done here.

36. Results, line 228: since the Greek symbol superscript was used for Delta on line 230, for consistency, the Greek symbol for Alpha should be used here vs ‘A’.

37. Results, line 232: is the 43% seasonality fluctuation defined based on a reference(s)? For December 21, is the year 2021?

38. Results, line 238: it would be helpful to clarify how adjustments were made for weather, i.e., contact increases due to colder weather where individuals tend to remain indoors more, even if this is further described in S1 of the appendix.

39. Results, lines 240-241: include the type of virus for each vaccine product.

40. Results, line 242: was the vaccine given to 60+ (the oldest age group) or was priority in Germany first targeted at even more at risk, e.g., 80+ or those with comorbidities or in long-term care in Germany? If this differs to the historical real-life strategy, state how this might have affected the pre-simulation fitting, etc.

41. Results, line 259: have ‘U, V’, Im, PI, NI’ been previously defined?

42. Results, line 268: can a reference for the dynamics in Germany be cited?

43. Fig 3 caption: define ‘E, P, I, L’.

44. Results, line 273: remove the duplicate ‘the’.

45. Results, 279: it would be useful to state what the seasonal fluctuation were. Late August 2021 was unusually cold, but what was the weather pattern in August 2020?

46. Results, 281: perhaps instead note end of December 2021 during the Christmas holiday, and make it explicit that this peak occurred at the end of December because thereafter stricter measures where instated.

47. Results, line 288: this should be ‘increased again as of mid-October’.

48. Results, lines 290-291: state why infections among children and teens were higher in the 3rd and 4th waves vs 1st and 2nd waves, e.g., because of vaccine eligibility? Similarly for the incidence among <5 in the 4th wave vs teens.

49. Results, lines 296-297: is there a published strategy (press release, report) than can be cited for Germany?

50. Results, line 297: did vaccination rates increase in only the 60+ group here, or was eligibility already extended to younger groups (hard to discern as there is no time period specified)?

51. Results, line 300: here ‘Johnson & Johnson’ was used, but on lines 241 and 243, no spaces were used for Johnson&Johnson, make consistent.

52. Results, line 301: add ‘of’ before ‘2021’. This timing and coverage level should be stated earlier or this section referenced above (see comment above).

53. Results, line 304: reword as ‘during the 2021 flu season’

54. Results, line 307: correct the typo ‘form’ to ‘from’

55. Results, line 308: replace ‘hypothetical’ with ‘counterfactual’

56. Results, line 309: what period does ‘current’ represent, November 22, 2021 the start of the simulation period? And should it be ‘i.e., schools remained closed’

57. Results, line 311: on line 286, it is stated that a fourth wave started late August 2021, are the authors predicting that this wave will last until May 2022 (~8 months+) or am I misunderstanding?

58. Results, line 313: it would be useful to quantify the rough peak estimate for incidence, in addition to stating that it would be 7x-higher than for the 2nd wave. The same for this convention on lines 316-317.

59. Results, line 324: it is rather schooling from home (i.e., via Zoom by teachers) vs home schooling (which implies lessons are given by parent(s))

60. Results, line 333: add a comma after ‘approved’

61. Results, line 346: are the authors suggesting mandatory vaccination for individuals who reside in Germany? Would there still be an influence from immigration, or would anyone entering the country also have to be vaccinated?

62. Results, line 349: how relatively short is this period? What is the daily vaccination rate?

63. Results, lines 350-351: vaccination may be much less effective than school closures or an emergency brake from these modelled findings, but it is important to state that NPIs cannot be maintained indefinitely and there are economic and social implications, which can be avoided through vaccination.

64. Results, line 359: replace ‘mutation’ with ‘variant’, as there are >1 mutations, i.e., N501Y, A570D, etc.

65. Discussion, lines 367-368: this may have been true at the time of writing, but many governments have now relaxed measures – this should be reflected prior to publication.

66. Discussion, line 374: the wording ‘do not everywhere progress’ is awkward

67. Discussion, line 374: what is the definition for under-aged, not eligible for vaccination, i.e. <5?

68. Discussion, line 391: revise to ‘that will occur in the future’

69. Discussion, line 392: are individual mutants captured in the model or new variants of concern?

70. Discussion, lines 400-401: how do (1) effectiveness and (2) protectiveness from infection and transmission differ?

71. Discussion, line 402: revise to ‘not all vaccines are equally’ and close brackets after reference 13, and the last part of this sentence does not make sense, do the authors mean ‘do not fully protect’?

72. Discussion, lines 403-406: it seems a bit out of scope for a modelling paper to discuss why vaccines do not fully protect. If this is retained, neutralizing antibodies, memory cells, CD4 T cells, etc. This is complex and not fully understood.

73. Discussion, line 458: insert ‘cold weather’ in front of ‘extremes’

74. Discussion, lines 467, 469, 471: define to what ‘currently’ refers

75. Discussion, line 469: consider adding ‘and subregion’ after ‘to any country’

Reviewer #2: Predicting the impact of COVID-19 vaccination campaigns - a flexible age-dependent, spatially-stratified predictive model, accounting for multiple viral variants and vaccines

The subject is interesting, but the authors must correct and explain these points:

Title

1. The title is not informative. Consider revising it.

Abstract

1. Background: The study question is not mentioned, and the study's rationale is obscure. Please make the objective of the study crystal clear, using a sentence beginning with "the objective of this study was to …".

2. Conclusion: Considering that the title presents the study question and the conclusion presents the answer the authors found for that question, the conclusion needs to be re-written. The conclusion needs to be inevitably derived from the results section.

Introduction

1. A paragraph is missing between lines 51-52 to present the existing gap in the literature.

Methods

1. Please include an overview at the beginning of the methods section.

Discussion

1. Please include the model's implications at the beginning of the discussion section.

6. PLOS authors have the option to publish the peer review history of their article (what does this mean?). If published, this will include your full peer review and any attached files.

Reviewer #1: No

Reviewer #2: No

---

## [Editor Report · Decision Letter 1]

31 Oct 2022

A flexible age-dependent, spatially-stratified predictive model for the spread of COVID-19, accounting for multiple viral variants and vaccines

PONE-D-21-37770R1

Dear Dr. Schneider,

We’re pleased to inform you that your manuscript has been judged scientifically suitable for publication and will be formally accepted for publication once it meets all outstanding technical requirements.

Kind regards,

Mohammad-Reza Malekpour

Academic Editor

PLOS ONE

---

## [Editor Report · Acceptance letter]

11 Jan 2023

PONE-D-21-37770R1 

A flexible age-dependent, spatially-stratified predictive model for the spread of COVID-19, accounting for multiple viral variants and vaccines 

Dear Dr. Schneider:

I'm pleased to inform you that your manuscript has been deemed suitable for publication in PLOS ONE. Congratulations! Your manuscript is now with our production department. 

Kind regards, 

on behalf of

Dr. Mohammad-Reza Malekpour 

Academic Editor

PLOS ONE